# Development of Rooftop Solar under Netbilling in Chile: Analysis of Main Barriers from Project Developers' Perspectives

Shahriyar Nasirov [1],*  , Paula Gonzalez [1], Jose Opazo [2] and Carlos Silva [1]

[1]   Faculty of Engineering and Sciences, Universidad Adolfo Ibañez, Avenida Diagonal Las Torres 2640, Peñalolén, Santiago 7941169, Chile
[2]   Business School, Universidad Adolfo Ibañez, Avenida Diagonal Las Torres 2640, Peñalolén, Santiago 7941169, Chile
*   Correspondence: shahriyar.nasirov@uai.cl; Tel.: +56-2-2331-1777

**Abstract:** The development of rooftop solar PV generation has significant potential to generate enormous benefits to the electricity systems in achieving emission reduction targets and meeting increasing global energy demand, but could also make the power systems more resilient and affordable. In 2012, the Chilean government introduced a net billing law (Law 20.517) to incentivize consumers to sell their excess renewable electricity into the grid, which was expected to lead to a significant growth in rooftop solar. However, to date, the advancement of these technologies in the country has been very limited due to various barriers. For this reason, identifying and mitigating the main barriers that impede the advancement of development of rooftop solar is necessary to allow the successful deployment of these technologies. Based on data collected from a questionnaire survey and interviews conducted among the project developers in rooftop solar, the authors identify and rank the major barriers to the adoption of these technologies in Chile. Our findings show that the most significant barriers include "high investment and recovery period for the customer", "lack of incentives to develop projects in the sector", "rigid regulations regarding project size", and "long administrative process and grid connection costs". Furthermore, we discuss the most critical barriers in detail together with policy recommendations to overcome them.

**Keywords:** rooftop solar PV; net billing; barriers; Chile

## 1. Introduction

Amid growing concerns for climate change and dependence on fossil fuels, the deployment of rooftop solar energy generation has become a crucial component of sustainable energy policies in many countries across the world [1]. The production of energy from rooftop generation may play a key role in achieving emissions targets and reducing electrical supply costs, decreasing electricity losses, improving quality of service, reducing line congestion, bringing communities closer to energy generation, and others. The development of technologies complementary to rooftop generation, such as economically attractive battery storage systems and the growing demand for electric vehicles (EV) may further accelerate their applications in the near future. However, the expansion of rooftop solar generation makes large changes to traditional electricity systems, creating opportunities and many challenges for energy markets, including operators, regulators, generators, new participants, grids, and a country's economy. In the new energy model, consumers become active prosumers who can generate, store, and distribute energy when necessary, that is, they can actively manage their own energy resources. Despite the important potential benefits of rooftop solar energy generation, the expansion of these applications compared to utility scale solar PV in the world countries, particularly developing countries, has been limited due to existence of several critical barriers. Among the critical barriers, institutional and regulatory barriers, economic and financial barriers, technical and infrastructure barriers, and public awareness and information barriers are the most studied in the literature [2–8].

Chile has taken decisive action plans to reduce their greenhouse gas (GHG) emissions significantly. Among the latest action plans, the Chilean government officially submitted the country's long-term low emissions strategy in the COP26 summit. The strategy included a long-term roadmap detailing specific sectoral objectives and goals to become carbon neutral and climate resilient by 2050. Thanks to several factors such as exceptional solar resources, attractive market conditions, and successful public policies, Chile has become one of the largest markets for investment in large-scale renewable energy projects in South America. In 2020 and for the first time ever, direct foreign investment (DFI) in the renewable energies sector represented nearly half of all investments of that kind in the country. At a systemic level, the expansion of renewable technologies (mainly solar and wind power) in total installed capacity has advanced faster than expected, reaching nearly 30% of the energy mix in 2021 [9]. However, it is important to note that the main focus of these investments has been in utility-scale projects and that small-scale rooftop generation played a limited role, corresponding to only 108 MW in February 2022, that is, 0.3% of total installed capacity at the national level [10]. The slow progress shown by rooftop generation reflects the possible presence of barriers that hinder a greater adoption and market growth. In the literature, there exist significant number of studies that aimed to study the barriers for the development of the rooftop solar PV adoption; however, most of these investigations have been undertaken in developed countries such as the US and Germany, and China, a case in itself [11]. In the case of Latin America, few studies have been carried out regarding this area; particularly in Chile, most of the studies have focused on the analysis of large-scale photovoltaic generation and the few studies of rooftop generation have focused on evaluations of economic and financial aspects of the projects [12,13]. Consequently, there is still a lack of studies that analyze the main barriers in a comprehensive manner. Among the limited studies considering solar rooftop projects in Chile, it is important to mention the research of Martínez [12], who evaluated the main economic factors that affect the adoption of photovoltaic net billing systems in Chile. The evidence provided by the analysis shows that, through different business models, it is possible to reduce some of the barriers faced by consumers, and that the type of business model used along with local regulation considerably influences the costs that customers face. In another study, and more recently, Aravena et al. [13] examined the barriers and opportunities in the deployment of the photovoltaic prosumer segment in Chile. For these purposes, they use a quantitative methodology focused on economic and financial aspects, whose three main indicators are the simple pay-back period, the internal rate of return (IRR), and the levelized cost of electricity (LCOE). Based on the measurement of these indicators, they conclude that the current regulatory framework is insufficient to be able to develop the full potential of this segment, with the main barriers being high investment costs and low household income. For this reason, greater financing options, regulatory changes, and appropriate promotional energy policies can speed up deployment without significantly affecting public spending for this purpose.

The present study fills this gap and provides empirical and analytical evidence for analyzing the main barriers affecting the implementation of rooftop solar power projects under net billing from the developers' perspective in Chile. In addition, another important objective of this study is to offer a discussion on the most ranked barriers to rooftop solar power applications in the country. Based on the findings, face to face interview results, and descriptions of the current discussions in the literature, this work then discusses key mitigation areas for consideration with regard to the transition pathways toward rooftop solar expansions in the Chilean energy system. The methodology used in the study is based on quantitative (questionnaire survey) and qualitative data collection (a series of semi-structured interviews). The study is structured as follows: Section 2 provides a literature review on business models for rooftop solar generation; Section 3 describes the Chilean electricity market and net billing regulation framework; Section 4 presents the research methodology; Section 5 outlines results and discussions; and finally, Section 6 concludes the paper.

## 2. Literature Review on Business Models for Rooftop Solar Generation

Different business models for rooftop solar PV started to gain interest during the last decade with the innovation and growing competitiveness of renewable energy technologies [14]. The business models developed for this sector in each country are different, but have some elements in common. Table 1 describes some of the most common models in international literature.

**Table 1.** Characteristics of distributed energy service business models.

| Business Model | Equipment Owner | Segment Applied | Financing System | Location |
|---|---|---|---|---|
| Consumer ownership | Consumer | Owner with access to capital and/or low interest rate | Own capital or debt | Consumer real estate property |
| ESCO (energy service company) ownership | ESCO, with right to buy at end of contract | Consumer with low access to capital | ESCO-financed | Consumer real estate property |
| Community-owned | Consumer and/ or investor | Communities | Own capital/ debt/third-party capital/Donations | Virtual plants |
| Cross-selling | Property owner | Small and medium-sized business owners | Property owner | Consumer ownership |
| Roof rental | Third parties, with right to buy at end of contract | Property owner or renter | Third parties | Consumer real estate property |

### 2.1. Consumer Ownership Models

The consumer ownership model in turnkey projects is one of the business models most developed around the world. In this system, the consumer is owner of the solar PV system and the main person responsible for the electricity produced by the project [15–21]. This model has been mostly implemented in European countries thanks to the advantage of accessing low transactions costs with permits and some purchase subsidies [12]. Under this model, the consumer ultimately funds the entire solar PV system and finds its own service provider depending on the service. These mainly include the distribution of solar PV systems, consulting on the design, sizing, and roof studies, and comprehensive services that combine the installation of equipment, monitoring, repair, warranties, and replacement parts. In this model, consumers benefit from their own electricity savings and excess energy billed to the grid under net metering and net billing systems. Under net billing systems for customers or end users, the excess energy is valued at the distribution company's average rate, that is, the same purchase price for customers; therefore, it is only valued in terms of energy and not power and they only receive a discount for this amount in their electricity bill. In many cases, the credit is paid at the level of the wholesale electricity price. In contrary, under net-metering, energy injected into the grid is paid at the complete retail rate [22]. Figure 1 shows the details of the consumer ownership business model.

In practice, there are three variations of the consumer ownership model that are mostly applied, and these vary by the level of company involvement in the technology adoption process. The first is the basic model where the integrator/installer sells the system to the owner and installs it with some additional services including O&M and system monitoring. The owner is responsible for all other aspects of system installation, including financing, interconnection, permits, and inspections [23]. In the second variation of the consumer ownership model, the integration/installation company assumes other parts of the installation process to simplify it for the customer. These include interaction with the distribution company for interconnection/net billing, assistance in the inspection, etc. The owners still pay for the entire system and are responsible for arranging financing. The third variation is the one-stop model. Under this model, the integrator/installer provides the customer with financing options already established with the banks or financial entities. The main barrier for the consumer ownership model is the initial investment required.

Evidently, the disadvantages to the consumer ownership model include high start-up and maintenance costs, transactions costs associated with grid connection, and the risk of poor system performance [17]. The consumer ownership model in Chile, where there are no subsidies for the supply or demand of rooftop generation systems, has a high initial cost and a recovery period of at least 6 years [24]. For this reason, the main customers are those who feel a special call to protect the environment and have high purchasing power, since the decision is more environmental than financial. These are generally rooftop customers, but others include small and medium-sized companies and farmers.

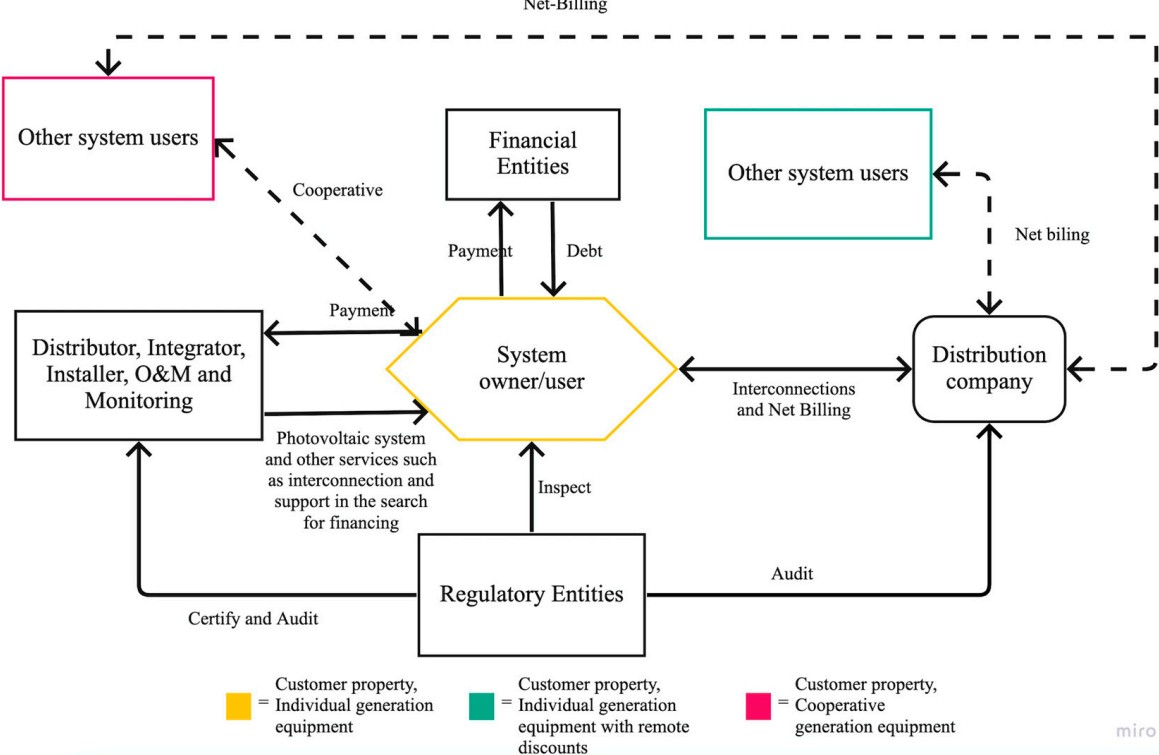

**Figure 1.** Consumer ownership business model, Source: Own Elaboration.

## 2.2. Third-Party Ownership Models-ESCO Business Model

ESCO-type third-party ownership models started appearing in the U.S. in 2005 and were initially applied to commercial or institutional end users. Since 2010, they have also seen rapid growth in the rooftop sector [17]. In this model, the PV system is owned by a third party, who is responsible for the installation, engineering, maintenance, and financing services for the system located on a host customer's property (See Figure 2). The system's owner (third-party) signs a solar lease agreement or a PPA with a host customer for 10 to 25 years. In the case of a lease agreement, the customer pays a fixed monthly rent payment regardless of the real amount of energy produced, while under a PPA, the customer pays an energy bill based on generation. This model plays an increasingly important role by addressing the high initial costs and other barriers that arise [15]. The goal of an ESCO company is to profit from investments made through the sale of energy produced by the users' systems, typically at a rate lower than retail price. Under this model, the ESCO owns and is responsible for the equipment and electricity production. Therefore, the consumer practically takes no investment risks or technical operations risks. There are two variations of the ESCO model. In the first, the ESCO supplies the energy and the customer purchases the energy at a lower price than it would pay to the grid. Generally, these PPA-type contracts are long-term, between 10 and 15 years, with a price per kWh fixed by the parties, while the ESCO is responsible for the investment, studies, installation, and maintenance of the generation equipment, so that the ESCO assumes all economic and technical risk.

In the second variation, the ESCO's revenue comes from the energy savings achieved by the consumer. Markets with preferential policies for net billing and interconnection, clear laws or regulations, and local financial incentives, generally favor the adoption of this model [17]. The net electricity bill savings generated by this model are between 10% and 20% [25].

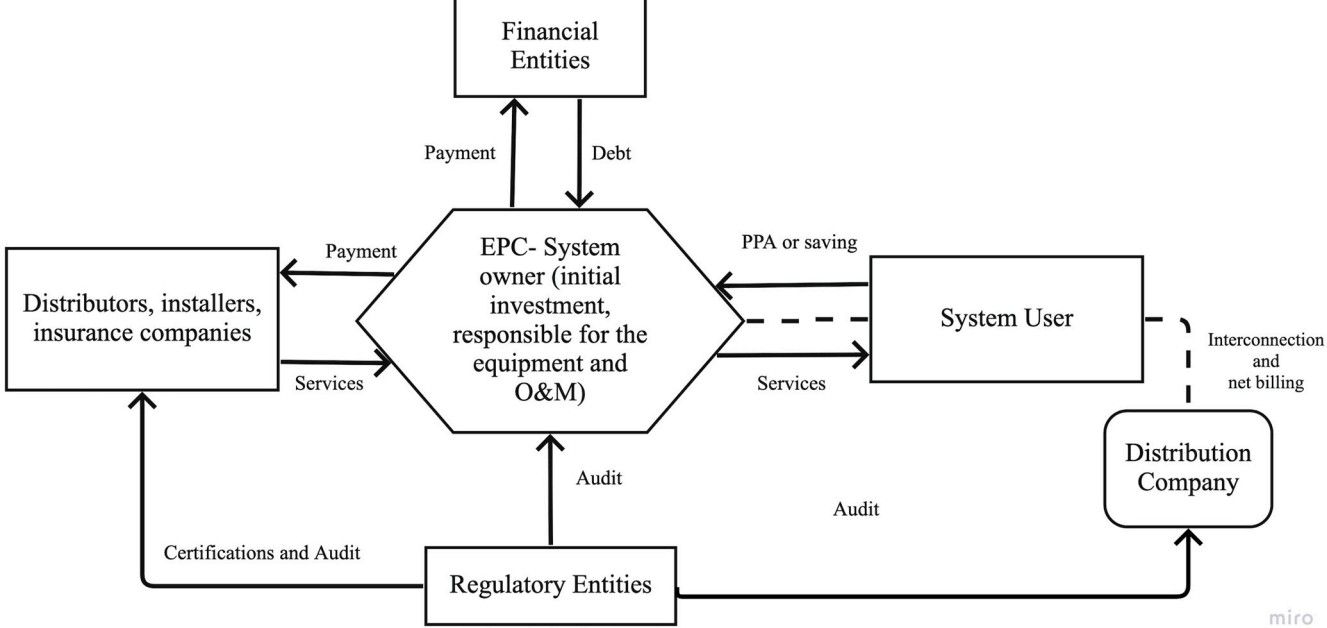

**Figure 2.** ESCO business model, Source: Own Elaboration.

*2.3. Other Ownership Models*

Among the other rooftop solar business models, cross-selling and the community model are two of the most common in more advanced countries. In cross-selling, the PV system is purchased along with a property, such as a house, apartment, or condo, therefore the price is added to the cost of the property, offering a solution that is designed harmoniously with the property's architecture, making it more aesthetically attractive than a complementary system [16]. This model is mostly found in Japan due to the large development of the prefabricated homes market and the short useful life of real estate properties, making it 10% cheaper for the customer than reconditioning systems [26]. The PV system expenses are generally incorporated into the home mortgage, which reduces transactions costs and interest rates, thus benefiting from the effects of anchoring. On the other hand, the community model is a system that accommodates multiple users, who often lack the adequate solar resources on-site or purchasing power or building ownership rights, to purchase part of their electricity from a solar installation located off-site [27]. Customers can sign up for these projects that have solar panels on farms or in gardens. Therefore, for the members of the community, the model provides a profitable alternative that allows them to use renewable energy through virtual net billing [15]. In this way, they can take advantage of economies of scale thanks to multiple investors, favoring a community or group of people who purchases a large-scale PV system for joint benefit [21].

**3. The Chilean Electricity Market and Regulatory Framework under Net Billing**

Chile is known for having one of the first market liberalization reforms initiated in the electricity sector. The Chilean electricity market is comprised of three main activities—generation, transmission, and distribution—with only the participation of privately-owned companies. The generation segment is a free market that has an installed capacity of 27 GW. Approximately 31% correspond to non-conventional renewable energies (19.55% solar PV, 0.36% concentrated solar, 0.17% geothermal, 2.04% small-scale hydropower, 1.61% biomass,

and 12.68% wind power [10]. According to the size and regulation, different types of companies participate in the generation segment by scale. These includes large generators that have more than 9 MW of capacity, small-medium distributed generation (PMGD) whose capacity surplus is lower or equal to 9 MW and is connected to the distribution networks, small-medium generators (PMG) whose surplus power available to the system is lower or equal to 9 MW and is connected to the transmission system, and finally, generators under net billing are renewable generation means of less than equal to 300 kW. The spot market in Chile was designed in such a way that only generators can participate. Therefore, generators purchase and sell energy under two types of markets: the spot market and the contract market. The spot market in Chile was designed in such a way that only generators can participate. Therefore, generators can trade their energy with their peers under the spot market or sell to free or regulated clients under the contract market.

The transmission market is a natural monopoly; therefore, the State regulates its activities through valuation, qualification, and rate-setting of its facilities. The transmission system expansion is planned by the National Electric Commission (CNE), and its works are subject to public bids organized by the Electricity System Coordinator. The transmission market is divided into four systems: national transmission, zonal transmission, transmission systems for development poles, and dedicated transmission. The planning process of each installation is carried out by the CNE every 4 years. Access to transmission facilities is open, except in dedicated systems, where their access is determined by the available technical capacity, which is calculated by the Coordinator. According to the latest Transmission Law, final consumers are responsible for the transmission fee.

The distribution segment is also considered a public service and natural monopoly, and its activity is regulated by the State and remunerated with rates applied to the final customers, based on an efficient model company similar to the yard-stick model. The remuneration is determined by the authority in four-year terms assuring profitability levels for the industry in the order from 6% to 14%. The remuneration of this segment is called Value Added Distribution (VAD), which considers investment costs, losses, operating costs, administration expenses, maintenance, customer service and billing, adaptations to demand, and is based on a model of efficient company. This process is carried out every four years by the CNE based on new cost evaluations. Both the distribution and transmission systems must consider future growth of PMGD, PMG, and net billing projects in the capacity planning in order to anticipate infrastructure requirements and avoid possible problems in the network, such as congestion.

According to the regulations on the Chilean electricity sector, there are two types of customers: regulated and free. Regulated customers are consumers with an installed capacity of less than or equal to 500 kW, and these customers are supplied through joint supply bids among all distributors. Customers between 500 kW and 5 MW can choose between the regulated or free modality. The net-billing law only applies to regulated customers and looks to promote the generation of one's own energy based on non-conventional renewable energies (ERNC) and efficient cogeneration. On the other hand, free customers participate in the contract market for the awarding of PPAs. Figure 3 provides a functional description of the different generation segments within the Chilean market.

In addition, there exist a wide range of institutions to establish the necessary conditions for industry expansion. The Ministry of Energy is the entity responsible for policies, plans, and standards for the development of the energy sector. Through this ministry, other authorities like the CNE are responsible for the planning of transmission, rates, and technical standards. Meanwhile, compliance with regulations is controlled by the Superintendency of Electricity and Fuels (SEC). Other industry players include distribution companies, who are responsible for ensuring the connection of generation equipment, planning or other works to the grid, evaluating the cost of equipment connection, and then calculating the discount or payment of excess energy generated according to law.

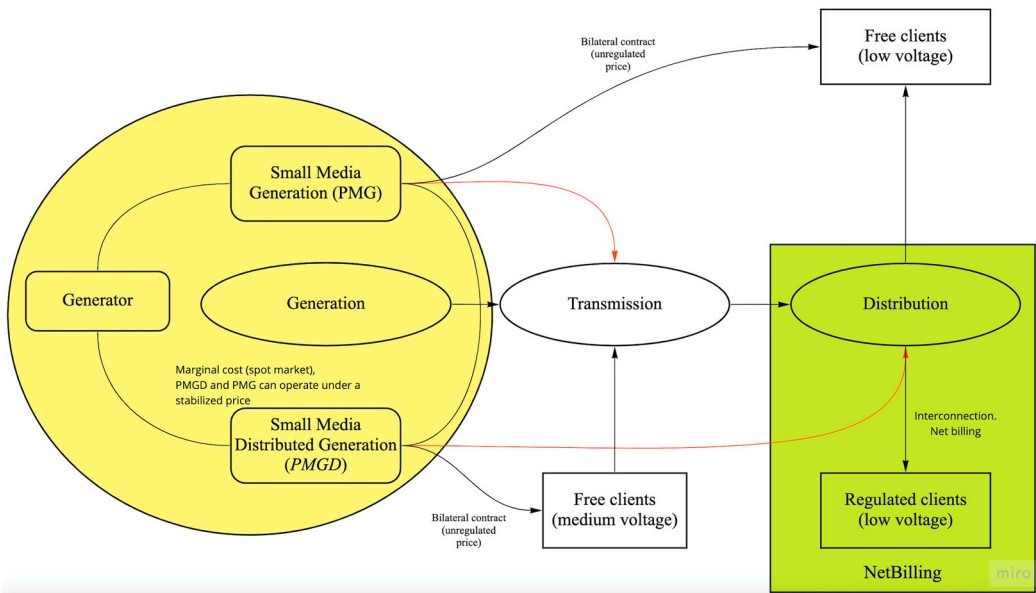

**Figure 3.** The structure of the Chilean electricity market for free and regulated customers.

*Regulatory Framework under the Netbilling Law*

The solar PV installed capacity under the net billing law reached 100.4 MW of capacity, corresponding to 9227 number of installations in 2021 (see Figures 4 and 5). As it can be seen the largest number of facilities are concentrated in the metropolitan regions with one of the highest installed capacities (28 MW, 26%) and the Atacama region with a low level of installed capacity (3.5 MW, 3%). Regarding the destination of these facilities, the majority of the projects correspond to rooftop projects, and with respect to the largest installed power per project, the majority is concentrated in agricultural projects with 33%, followed by industrial with 23%, and rooftop residential with 16%. This contrasts with the weak increase in rooftop generation which, seven years after the net billing law went into effect and despite its high potential and enormous benefits, shows highly limited progress, corresponding only to 108 MW in February 2022, that is, 0.3% of total installed capacity at the national level.

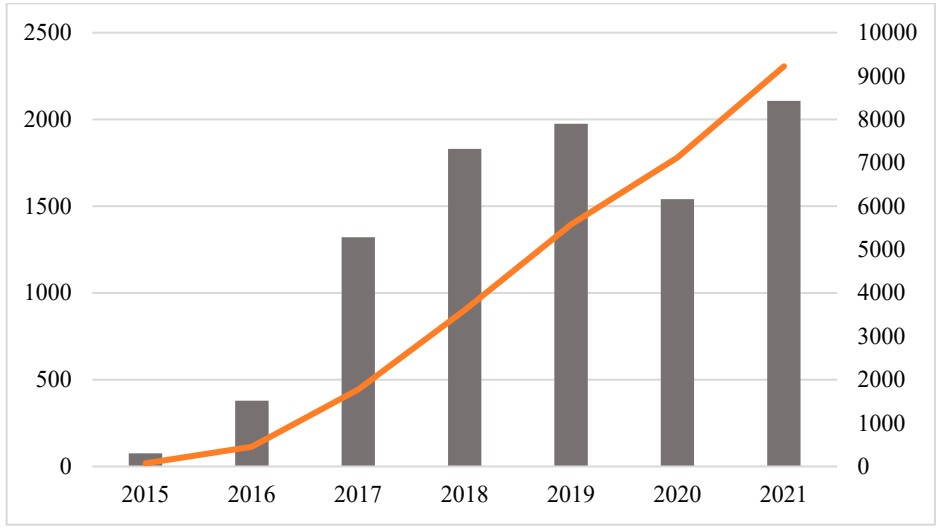

**Figure 4.** Number of Solar PV Installations under Net Billing in Chile. Source: Own elaboration based on data obtained from Energia Abierta [10].

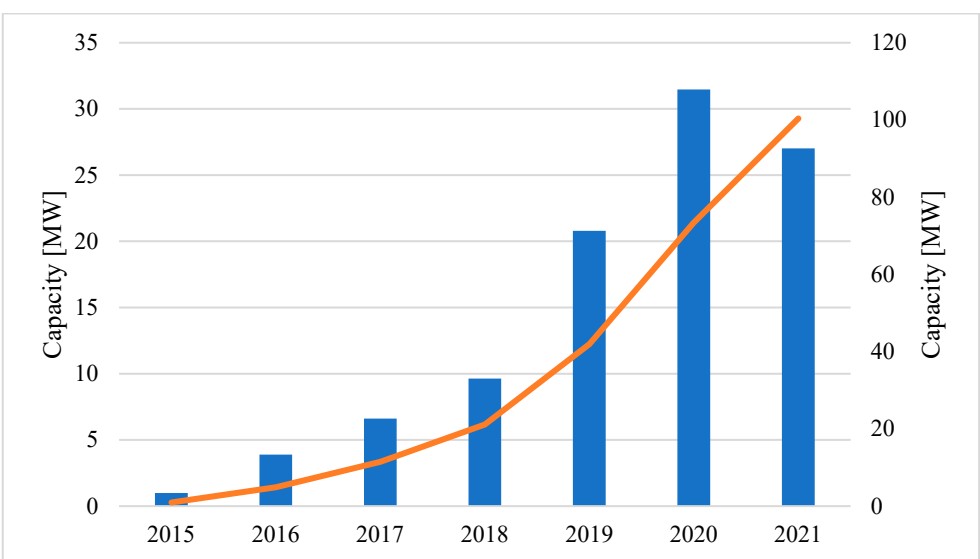

**Figure 5.** Solar PV installed capacity under net billing in Chile. Source: Own elaboration based on data obtained from Energia Abierta [10].

The first regulatory initiatives regarding self-generation date back to 2004, where self-generation projects were integrated into facilities for the sole purpose of consumption and regulated through Electrical Technical Standard No. 4 [28]. However, according to this regulation, these projects could not inject excess energy into the distribution grid because the standard required equipment to ensure compliance with this condition. The first steps towards what is now known as Netbilling started in 2008, when the standard was reformed to establish a discount and remuneration of electrical rates for rooftop generators. It was this bill that years later, specifically in 2012, would become Law 20,571, currently known as the "Netbilling Law," which modifies article No. 149 of the LGSE (LGSE: General Law of Electricity Services) [29].

Under the original Netbilling law, the electrical generation equipment projects using NCREs and efficient cogeneration facilities shall have a capacity of up to 100 kW, applicable only to regulated customers. This law allowed customers to benefit from their own electricity bill savings thanks to self-supply, while also benefiting from the excess energy generated through injection into the grid, valued at the same price at which the respective distribution company buys energy from the generation companies with which it holds supply contracts.

Over the years, expectations were not met regarding the number of customers who would implement distributed rooftop generation. Based on this and campaigns held by companies and associations within the sector, the Netbilling Law was modified by Law No. 21,118 of the Ministry of Energy [30]. The most important changes made were the following: the maximum limit for project development was increased from 100 to 300 kW; customers connected to the same concessions company with connections at different addresses can apply for coverage under the Netbilling Law to eliminate certain barriers related to resources and financing; and customers can discount charges for electrical supply from other facilities or properties owned by the same customer if energy injections have not been discounted during the respective term, and the revenue received for energy injection does not constitute income and is not subject to VAT. Finally, in 2019 and 2020, the Technical Standard on the Connection and Operation of Generation Equipment and the Regulations on Distributed Generation and Self-Consumption were passed, respectively. These regulations look to improve how the law operates, defining several standards on the interconnection procedure, including communications media, response times, and costs associated with the process, among others.

## 4. Methodology

The methodology used in this study is a mixed approach which is based on quantitative and qualitative data to analyze barriers from the perspective of different business models under net billing in Chile. In the literature, there are several methodology approaches to analyzing barriers, but most have used approaches from the perspective of different business models applied in different sectors [15–17,20]. During the first phase of this study, we collected quantitative data through surveys to obtain and elaborate data quickly and effectively [4–6,31–35], and the second phase included semi-structured interviews to discuss and explore more critical barriers. The surveys have advantages due to their less time consumption, larger scope, greater objectivity, easy comparison, and generalization. On the other hand, the interviews allowed for the collection of detailed responses, therefore performing interviews during the last phase provides more information regarding the most relevant barriers. The survey structure is comprised of the definition of the variables and sample, the questionnaire design, and data analysis. The structure of the methodology is presented in Figure 6.

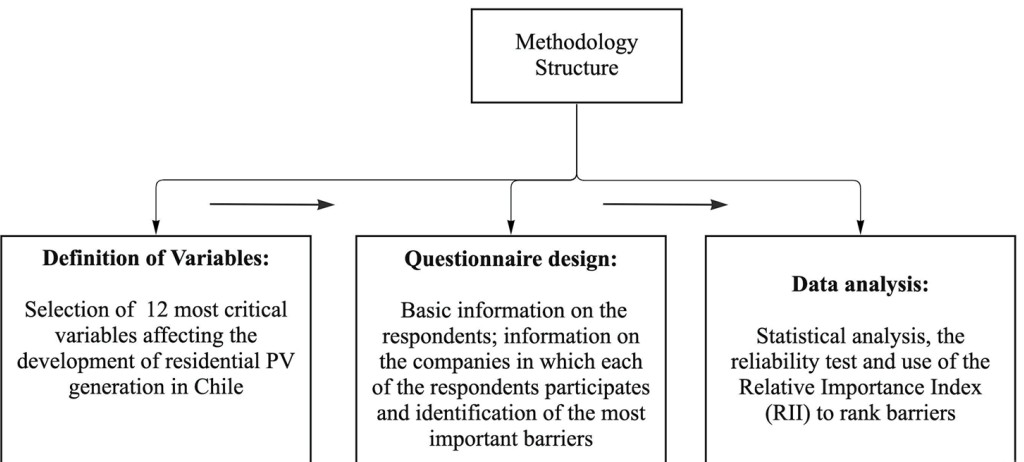

**Figure 6.** The structure of the methodology, Source: Own elaboration.

### 4.1. Definition of Variables and Sample

The barriers identified in the literature review provide extremely useful information for several countries. However, as mentioned before, these barriers may be very specific to a country or region. Therefore, to identify the most relevant barriers in Chile, the study initially considered the most common barriers identified in international literature and defined a preliminary list of barriers, considering the context of the Chilean PV market in net billing segment, information collected from journal publications, conference recordings, and documents from public and private institutions. This list was verified by a small pilot study aimed at establishing the solar rooftop PV barriers applicable to Chile, which addressed several professional experts with over 10 years of experience in the industry and who have held top positions in their organizations, making them a reliable information source. The experts' opinions and experiences were considered to determine the 12 most critical variables affecting the development of rooftop PV generation in Chile (see Table 2). The barriers are grouped into four categories: economic and financial; technology and market; regulatory policies; and social and environmental.

**Table 2.** Barriers affecting the implementation of solar PV projects distributed in Chile.

| Dimension | Netbilling Segment Barriers |
| --- | --- |
| Economic and financial | LCOE of PV compared to electrical grid price<br>Difficulty obtaining financing<br>High investment and recovery period for the customer |
| Technology and market | Lack of qualified labor and specialist companies<br>Grid structure, capacity, and regulation for expansion<br>Service provider companies have a hard time competing with distribution companies |
| Regulatory policies | Long administrative process and grid connection costs<br>Lack of policy incentives to develop projects in the sector<br>Rigid regulations regarding project size |
| Social and environmental | Final user conduct and behavior<br>Customer's lack of knowledge and access to information<br>Difficulty developing projects with associative systems |

The survey participants covered the expert professionals working to develop PV projects under net billing law, considering one representative per company. Since there is no database or list of all individuals within this population, the decision was made to apply a non-probabilistic sampling technique based on quota sampling to guarantee representativity and ensure collection of the necessary information for the statistical significance of the target groups [32–35]. One of the disadvantages of this methodology is that when defining the ranges, possible levels of the population may be left out, therefore the results of this study must be considered to be exploratory and not conclusive. To select the population, companies associated with the preliminary list of companies were sought out based on their business category, subcategory, and economic activity in the database kept by the Chilean Internal Tax Service (SII). Based on these characteristics, a new search was performed within the SII database for other companies with the same characteristics, identifying 62 companies that develop PV projects under net billing segment of the solar PV market.

*4.2. Questionnaire Design*

In this phase, the questionnaire was designed to obtain the most critical barriers. The questionnaire consists of three parts: the first explains the objectives of the study and aims to collect basic information on the respondents, including position, area, highest education level, etc., and the second part collects information on the services, projects, and business models of the companies in which each of the respondents participate. Finally, the third part looks to identify the most important barriers to the implementation of rooftop solar PV projects. For this, respondents were asked to quantify how relevant they considered each of the barriers identified to be on a 5-point Likert scale, where 1 = "extremely irrelevant", 2 = "fairly irrelevant", 3 = "somewhat relevant", 4 = "highly relevant", and 5 = "extremely relevant". In addition, respondents were given the option to mention other relevant barriers.

The questionnaire was sent to individuals through a link to provide their answers online. The survey process continued until the number of responses reached the sample size to obtain a representative sample and allow for coherent statistical analysis [36] with a confidence interval of 90% and a margin of error of 0.1% [33,35]. Finally, a total of 36 representatives from PV net billing development companies responded to the survey. Therefore, the total exceeded the determined sample size of 32 representatives.

The respondents in the sample have an average of 10 years of experience developing solar PV projects distributed in Chile. More specifically, as shown in Table 3 the respondents with 1 to 5 years of experience were only 23% of the sample, while 34% had between 6 and 10 years of experience, 19% between 11 and 15 years of experience, and 25% over 15 years of experience in the sector. In terms of their role in the industry, 58% of the respondents were managers, 11% directors, and the remaining 30% held other positions, like consultants

and specialists. Regarding education, 2% had graduated from a technical training center or professional institute, 45% had a university degree, 11% a postgraduate diploma, 38% a master's degree, and 4% a PhD. Finally, regarding the organization size, 19% belonged to a large company, 32% to medium-sized companies, 21% to small companies, and 28% to micro companies (Table 3).

**Table 3.** Main characteristics of survey respondents.

| Years of Relevant Professional Experience in the Sector: | Netbilling Projects |
|---|---|
| 6–10 | 35% |
| +15 | 26% |
| 1–5 | 24% |
| Education: | |
| College graduate | 50% |
| Master's degree | 21% |
| Postgraduate certificate | 6% |
| Job position in the organization: | |
| Manager | 59% |
| Other | 29% |
| Director | 12% |
| Organization size: | |
| Medium (25–100 employees) | 26% |
| Micro (0–9 employees) | 38% |
| Small (10–25 employees) | 21% |

*4.3. Data Analysis*

The data collected from the survey underwent statistical analysis using the statistical software SPSS, version 24. The reliability of the classifications was verified using the Cronbach's alpha value; according to Shen et al. [11], a Cronbach's alpha value of 0.7 or greater indicates a reliable set of group classifications. For the data collected in the survey related to the rooftop PV developer company segment, the Relative Importance Index (RII) was used. The RII calculates the mean value of each barrier and has been widely used in the literature for analyzing critical factors or barriers to renewable energies [33,37]. If two barriers have the same mean score, the highest range is assigned to the barrier with the lowest standard deviation. The main advantage of the RII method is its simplicity.

**5. Results and Discussion**

Table 4 summarizes the data analysis results obtained from the surveys on the most relevant barriers affecting the implementation of solar PV projects in the Netbilling segment. The results show that the most important barriers affecting the implementation of Netbilling PV projects are "High initial investment and recovery period for the customer" (E3), followed by "Lack of incentives to develop projects in the sector" (G2), then "Rigid regulations regarding project size" (G3), and "Long administrative process and grid connection costs" (G1). The Cronbach's alpha coefficient is 0.796, which is higher than 0.7; therefore the 5-point Likert scale used is reliable.

Figure 7 shows the relative percentage of each barrier and each category over the total. As can be seen, the technology and market barrier categories are the most relevant, followed by regulatory policies with 26%, economic and financial with 25%, and finally, social and environmental with 22%.

**Table 4.** Initial matrix of RII results by barrier for the implementation of PV Netbilling projects.

| Netbilling Barriers | RII | DE | Final Order |
|---|---|---|---|
| LCOE of PV compared to electrical grid price | 3.7 | 0.93 | 8 |
| Difficulty obtaining financing | 3.8 | 1.07 | 6 |
| High investment and recovery period for the customer | **4.0** | **1.03** | **1** |
| Lack of qualified labor and specialist companies | 2.9 | 1.28 | 12 |
| Grid structure, capacity, and regulation for expansion | 3.5 | 1.31 | 9 |
| Service provider companies have a hard time competing with distribution companies | 3.4 | 1.35 | 11 |
| Long administrative process and grid connection costs | **3.9** | **1.23** | **4** |
| Lack of policy incentives to develop projects in the sector | **4.0** | **1.27** | **2** |
| Rigid regulations regarding project size | **3.9** | **1.35** | **3** |
| Final user conduct and behavior | 3.6 | 1.13 | 10 |
| Customer's lack of knowledge and access to information | 3.4 | 1.04 | 5 |
| Difficulty developing projects with associative systems | 3.3 | 1.03 | 7 |
| Cronbach's Alpha | | | 0.796 |

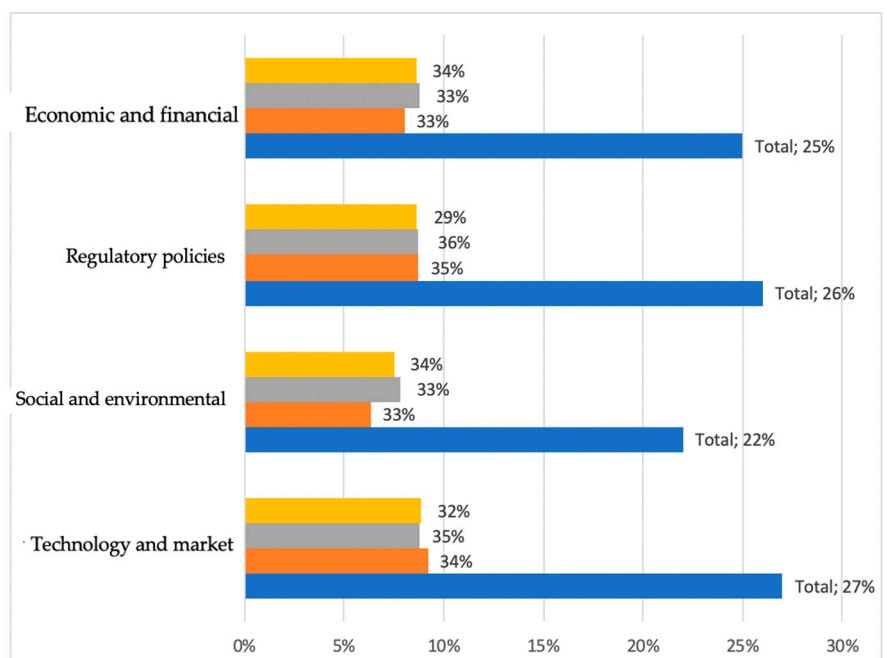

**Figure 7.** Ranking of categories and Netbilling barriers.

The following section discusses the four most critical barriers identified by data analysis:

*5.1. High Investment and Recovery Period for the Customer*

High investment and recovery period for the customer is the first barrier according to the ranking, which is consistent with the literature review and the barriers cited for the adoption of PV solar power in the context of many countries [3,38,39]. In this context, one of the important factors is related to the price of PV systems and its associated payback period and the purchasing power of the small-scale investors. Even though the price of small-scale PV systems in Chile has gone down recently, its cost continues to be higher in comparison to countries with more developed rooftop PV systems. According to data by the Ministry of Energy, for an installed capacity of approximately 1 kW, the average investment cost is around $2,200,000 Chilean pesos (US$2600) and the recovery period is 6 years [40]. The second important issue is associated with Chilean families' purchasing power. When comparing these figures to countries with a high-level of PV development in this segment, for example Germany, the prices in Chile continue to be very high, especially

for the rooftop segment and its socioeconomic level, where 68% of Chilean households earn less than the national average (US$1750), and national average household income is 33% lower than the average investment cost to install 1 kWp for a rooftop PV system.

Additionally, there are very few options for financing by the national banking sector, and potential users either have no knowledge of or no access to good financing alternatives. The segments most affected are rooftop and commercial customers who have access to limited green credit options and must finance the projects using consumer loans with high interest rates around 21% [41], negatively affecting the project's profitability, since when compared to electricity prices of around 0.16 USD/kWh, the LCOE averages over 0.26 USD/kWh for a system of 1 kW located in Santiago and generates 1494 kWh per year. Most financing options come from large commercial banks with no competitive conditions and there are limited attractive financing options from decentralized finance institutions with attractive conditions.

The number and magnitude of the new projects have important potential effects on the market at different levels (competitors, providers, transmission capacity, and regulation) and in different dimensions (financing, profitability, input prices, and construction). In practice, the viability and profitability of renewable projects today critically depend on the distributors and there is little transparency and long delays in the process to allow the entry of rooftop solar generation. For local grid companies, the entry of new generators reduces their electricity revenues and increases administrative costs, which is why they have incentives to postpone the entry and prevent the development of rooftop solar generation [42,43]. As a result of all this, there is a higher level of uncertainty, which has a negative impact on the cost and complexity of the connection process, reaching the limit of preventing the implementation of a project, even if it is profitable at a private and social level [33].

There are also several indirect costs or adoption costs that make the project less attractive in economic terms. Among the adoption costs, the soft costs (e.g., sales and marketing costs, installation labor, system design, interconnection, permits, and overhead expenses) are a large limiting factor for PV adoption. Martínez et al. [12] demonstrated that these costs may represent up to 60% of the total cost of a PV system, when up to 30% of these could be avoided.

*5.2. Lack of Policy Incentives to Develop Projects in the Sector*

The second most relevant barrier shown by the study results is the lack of incentives to develop projects in the sector. For most people, it is very hard to invest without substantial government support. Currently, there are different programs promoted by several public entities, such as the Casa Solar program by the Ministry of Energy, the Intraproperty Irrigation and Drainage Program (PRI) by the Agricultural Development Instituted (INDAP), among others. These programs look to provide incentives for adopting PV and other NCRE generation sources, financing up to 50% of the cost. Many of these include training, technical advisory, and energy education. However, these measures have still not been sufficient for encouraging the adoption of such systems. The main reasons for this include the low level of citizen awareness of the benefits of the energy transition; therefore, better education and information on renewable energy sources such as PV are crucial [43]. The Chilean government recently implemented an alternative for Netbilling customers to be able to collectively adopt PV systems, thus helping to reduce initial costs by up to 30%. However, there have been great difficulties in treating, negotiating, and reaching agreements among all parties involved, which generally increases costs and implementation periods. Another relevant barrier to the expansion of the PV Netbilling segment is the lack of storage systems, which are extremely expensive in comparison to other countries. Chile currently provides no policy incentives for distributed storage systems, which, compounded with lower socioeconomic levels, further hinders the adoption of these technologies.

### 5.3. Rigid Regulations Regarding Project Size

The third most important barrier according to results are the rigid regulations regarding project size. From a national perspective, this barrier is mainly due to the law (Law No. 21,118 and No. 20,571) on self-consumption. The previous Netbilling regulations considered a maximum capacity limit of 100 kW, but due to the low development of projects in the sector, changes were made to regulations, increasing the limit to 300 kW. This new limit was defined considering the necessary roof space required for the installation of a self-supply PV project in commercial and industrial companies (2000 m$^2$). However, this change in regulations was not sufficient, as there are still many segments left out of this regulation, mainly regulated customers between 300 kW and 500 kW, and projects that do not inject into the grid, but are not profitable according to the defined size. Therefore, these types of restrictions and demands make projects even more expensive and act directly as a barrier to entry, because the goal of this regulation is Netbilling, that is, benefiting from self-supply, reducing the electricity bill in terms of energy prices that consider energy and power, and benefiting from the payment of excess energy injected into the grid, when the balance between injections and consumption is positive for the customer. It should also be noted that these injections are not paid at the same rate as self-supply energy, but only in terms of the distribution company's energy purchase price, which is approximately 40–60% of this charge. To define the best alternative, other countries with a large installed capacity of distributed solar PV could be looked to as a reference. For example, in Italy, over 70% of the total solar PV capacity comes from distributed installations with no restriction to system size or grid injection amounts [42]. In the case of Germany, there are no restrictions on the installed capacity allowed, but there is a difference in the FIT based on the installed capacity, among other factors. The higher the installed capacity, the lower the FIT received by the customer. One of Germany's main motivations for introducing the FIT logic is to increase the demand for PV systems and, with it, reduce technology costs. As this occurred over time, the amounts associated with the FITs dropped.

### 5.4. Long Administrative Process and Grid Connection Costs

The fourth most relevant barrier according to the survey results is the long administrative process and grid connection costs. Netbilling projects face obstacles to implementation due to the long amount of time to connect them to the grid. In most cases, the developer companies perform this procedure due to information barriers and a lack of knowledge among potential users, mainly homeowners. In the case of delays, these customers can be discouraged and even abandon the projects. According to the people interviewed, these delays may be due to non-compliance or processing errors by the developer companies or by the distribution companies. In the case of developer companies, delays are generated mostly due to problems with the switchboards, conductor types, labeling, and others. Secondly, they may be caused by rejections related to the point of connection, and finally by rejections due to inconsistencies between the application and information forms, for example, with respect to the blueprint or land. In the case of distribution companies, the growth of the rooftop segment creates conflicts of interest with distribution companies, which have no incentive to improve grid connection times [43]. In terms of connection times in the best-case scenarios, not considering any discrepancies that might exist between the customer and the distributor, this could take around 9 months, according to the Technical Standard on Connection. In all cases, this process includes the formalization of a contract with certain guidelines according to regulations, but ultimately designed by the distribution company.

Other problems mentioned after the system connection consist of correct measurement and billing of injections and the balance by distribution companies. With respect to this issue, the regulator (The SEC) is implementing audits of the distribution companies' billing processes to improve their knowledge and perform correct measurement, since this process is fundamental to ensuring profitability and return on investment. According to current regulations, all information must be available to the customer for determining the size of

its system and estimating the related costs. However, in practice, this is not the case, with most customers having to request this information from the distribution company.

*5.5. Mitigation Measures for Main Barriers*

This section is complemented by a literature review and extended opinions and experiences of the experts interviewed over the mitigation of the main barriers. Table 5 summarizes the proposed mitigation measures for each main barrier.

**Table 5.** Mitigation Measures.

| Barriers | Mitigation Measures |
|---|---|
| High investment and recovery period for the customer | <ul><li>Tax incentives for purchase of solar systems;</li><li>Capital subsidies to low resource families for compensating a part of initial investment;</li><li>Financial incentives to include storage facilities;</li><li>Efficient options to reduce the soft costs.</li></ul> |
| Lack of policy incentives to develop projects in the sector | <ul><li>State programs to increase citizen awareness about the benefits of the solar systems;</li><li>Policy incentive mechanisms for bilateral electricity markets for buying and selling energy;</li><li>State programs to incentivize rooftop solar PV systems in new buildings.</li></ul> |
| Rigid regulations regarding project size | <ul><li>Elimination of the 300 kW limit established by law;</li><li>Requirement of a study about the technical and operational limitations of the grid.</li></ul> |
| Long administrative process and grid connection costs | <ul><li>Implementation of incentive schemes for distribution companies;</li><li>Creation of a new figure—demand aggregators—which can improve the relationship with the prosumer and the distribution company.</li></ul> |

## 6. Conclusions

During recent years, Chile has become one of the most attractive countries for the development of solar power projects. Thanks to its massive growth, solar power is now responsible for 19.55% of the energy matrix in Chile. However, nearly all the generation, around 98%, comes from large-, small-, and medium-scale generators, while only 2% belongs to rooftop solar projects, known as Netbilling. This low percentage held by the latter segment is because it is currently being developed and is facing some obstacles that hinder its progress. Due to the high resource potential and multiple benefits provided by this type of generation, including lower emissions, reduced electricity losses, and others, it is essential to identify the barriers currently faced by the segment in order to promote greater development.

Within this context, this study identifies and analyzes the main barriers affecting the implementation of rooftop solar PV generation projects from the perspective of developers through the collection of data through surveys and interviews. The study analyzes the data results using the RII methodology to rank the barriers, and then performs face-to-face interviews with experts. The results indicate that "High initial investment and recovery period for the customer", "Lack of incentives to develop projects in the sector", "Rigid regulations regarding project size", and "Long administrative process and grid connection costs" are the most critical barriers to the implementation of rooftop PV projects.

Since one of the major limiting factors has to do with the high initial costs compared to average Chilean household income, it is very important to create incentives such as financial assistance programs, or even co-financing, depending on the socioeconomic level of potential users, and others. Moreover, quoting platforms could be developed to reduce

soft costs, which are the highest costs faced by rooftop users. These platforms would help reduce search costs and increase customers' access to estimates, which should lead to customers receiving lower-priced offers. In practice, the applications show that PV prices are significantly lower on these active platforms than prices obtained directly. Another potential benefit of platforms is for monitoring the quality and reliability of installers. While the regulations allow only certified installers to build and connect these systems, these platforms help follow up on the installers and their services to verify that they are complying with current regulations. According to the results of this survey, 70% of the survey participants' companies do not offer financing; therefore, it is crucial to provide potential customers with the available financing information to promote adoption and improve their knowledge. In addition, current regulations limit part of potential rooftop solar power users, and those interviewed recommend eliminating the installed capacity limit or applying limits to grid injections, or establishing new conditions so that the projects are actually used for own supply and other potential adopters do not take advantage of the benefits of this law, since the remuneration rate is generally much higher than for other generation segments. This change might not only increase the number of users, but also improve the profitability of future projects, as well as recovery times, to mitigate these barriers.

**Author Contributions:** Conceptualization, S.N.; Methodology, S.N., P.G. and J.O.; Validation, C.S.; Formal analysis, P.G. and J.O.; Investigation, S.N., P.G. and J.O.; Resources, P.G.; Data curation, P.G.; Writing—original draft, S.N.; Writing—review & editing, S.N., J.O. and C.S. All authors have read and agreed to the published version of the manuscript.

**Funding:** This research was supported in Chile by the project ANID/FONDAP/15110019 (SERC-CHILE) and by the project CONICYT/Fondecyt Regular Nº 1180115.

**Institutional Review Board Statement:** Not applicable.

**Informed Consent Statement:** Not applicable.

**Data Availability Statement:** Not applicable.

**Conflicts of Interest:** The authors declare no conflict of interest.

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
