# Peer review of "Development of Rooftop Solar under Netbilling in Chile: Analysis of Main Barriers from Project Developers’ Perspectives"

_sustainability, doi:10.3390/su15032233_

Round 1
Reviewer 1 Report
Dear Authors,
The paper is well written. It presents an interesting take on problems facing rooftop PV deployment in developing countries, specifically Chile. I just have some minor comments:
1. Fig. 1 has some Spanish terms. These should be in English.
2. ESCO should be defined earlier than in line 155
3. let or net on line 236?
Author Response
Reviewer 1:
The paper is well written. It presents an interesting take on problems facing rooftop PV deployment in developing countries, specifically Chile. I just have some minor comments:
Response: Your recognition of our work is much appreciated. Many thanks for reviewing our work. Your suggestions are included in the main text.
Fig. 1 has some Spanish terms. These should be in English.
It is corrected.
ESCO should be defined earlier than in line 155.
It is corrected.
let or net on line 236?
It is corrected. Many thanks
Reviewer 2 Report
1) Authors should address about COP 26 in the present study
2) Author can refer few related articles: https://doi.org/10.1016/j.energy.2022.126415
Experimental assessment of thermoelectric cooling on the efficiency of PV module
3) The objective should address well at the last section of the article
4) Why the present study is focused on Chile? is there any special reason?
5) Table 2: Barriers section why codes are important, otherwise remove it. As a international reader a little bit difficult understand what does code refers?
6) The title of the article is main barriers and prospectives. Howerver, a table 2 does not justify to the title a deep investigation need to be conducted for Instance refer to following DOI'S: https://doi.org/10.3390/inventions7030053; https://doi.org/10.3390/app122010400 authors read and cite them. I don't agree a table without deep study. Moreover, the table represented by authors I feel they are recommendations.
7) Fig.7, please change your legends to English. Although, I recommend translation can be added in brackets. As a reader I don't understand what does the legends means
Author Response
Reviewer 2:
Authors should address about COP 26 in the present study
Many thanks for reviewing our work. Your suggestion about COP 26 was included in the Intrduction.
Author can refer few related articles: https://doi.org/10.1016/j.energy.2022.126415 Experimental assessment of thermoelectric cooling on the efficiency of PV module.
Thanks for your suggestion. The article was cited in the text.
The objective should address well at the last section of the article.
As suggested, we have clarified the objective of the study at the last section.
Why the present study is focused on Chile? is there any special reason?
We think it is an interesting country case study and can interest international readers. It is because Chile has an interesting energy system. It was the first country that liberalized its energy systems in the world. Unlike many countries, development of renewable energy sources is totally subsidy-free. The country has been converted into an energy hub. It ranks number one for investment climate for development of renewable resources. The topic is one of the focus topics in the renewable energy industry and finding of this study can also be applicable in many regional countries.
Table 2: Barriers section why codes are important, otherwise remove it. As a international reader a little bit difficult understand what does code refers?
Many thanks for your suggestion. We have removed them.
The title of the article is main barriers and prospectives. Howerver, a table 2 does not justify to the title a deep investigation need to be conducted for Instance refer to following DOI'S: https://doi.org/10.3390/inventions7030053; https://doi.org/10.3390/app122010400 authors read and cite them. I don't agree a table without deep study. Moreover, the table represented by authors I feel they are recommendations.
The title of the article is about Analysis of Main Barriers from Project Developers’ Perspectives. We do not focus on prospective. Table 2 include the most relevant barriers in Chile. The barriers identified in the literature review provide useful information for several countries. However, these barriers may be very specific to a country or region. Therefore, to identify the most relevant barriers in Chile, the study initially considered the most common barriers identified in international literature and this list was verified by a small pilot study aimed at establishing the rooftop PV barriers applicable to Chile.
Many thanks for your reference suggestions. We have included them in the study.
Fig.7, please change your legends to English. Although, I recommend translation can be added in brackets. As a reader I don't understand what does the legends means
Many thanks. It is corrected.
Reviewer 3 Report
Thank you for submitting your manuscript to Sustainability Journal.
Please find below some comments for your consideration.
Abstract: Line 15. Please add the word reduction. "....emission REDUCTION targets...."
Line 52: It seems contradictory when describing Chile as having " attractive market conditions and successful public policies", when you are identifying these points as barriers in the paper. Perhaps you would like to add "large-scale" in Line 56 - "......of all LARGE_SCALE investments....?". Kindly explain.
Line 70: "....he projects [9-10] consequently...." Please add a full-stop and capital C. "...he projects [9-10]. Consequently..."
Line 77: "In other study...." to change to "In another study...."
Line 93 has a different font size than the rest of the paper
Line 94: "onlines" to be replaced by "outlines"
Figure 1 should be fully translated into English language "ej? SEC, MINVU???" as well as the central box. Also please revise Figure 2 "ej??" Moreover, Figure 3 has some very very small text that cannot be read clearly. Please enlarge the font slightly.
Line 205: "...sale of energy..." to be replaced by "...sell energy..."
Line 215: "planification process" to be replaced by "planning process"
Line 220-231: How did the current worldwide energy crisis impact this process? Is it still being re-evaluated every 4 years similar to the business-as-usual scenario?
Line 244: "exist" to be changed to "exists"
Line 324: "y or region, therefore, t". Please add a full stop and capital T. "y or region. Therefore, t"
Line 337: What do you mean by "sample universe"?
Lines 411-423 identify two separate issues. The first is the cost of the PV system and its associated payback period. The second is the purchasing power of the small-scale investors. These two factors need to be considered separately because they are mutually exclusive. Please revise the paragraph.
Comment on Results: The analysis of results and proposals for mitigation are dealt with within the same paragraphs. May I suggest that a Table is added summarising the proposed mitigation measures for each barrier, before the Conclusion.
Author Response
Reviewer 3:
Thank you for submitting your manuscript to Sustainability Journal. Please find below some comments for your consideration.
Response: Your recognition of our work is much appreciated. Many thanks for reviewing our work. Your suggestions are included in the main text.
Abstract: Line 15. Please add the word reduction. "....emission REDUCTION targets...."
It is corrected.
Line 52: It seems contradictory when describing Chile as having " attractive market conditions and successful public policies", when you are identifying these points as barriers in the paper. Perhaps you would like to add "large-scale" in Line 56 - "......of all LARGE_SCALE investments....?". Kindly explain.
It is explained. Many thanks.
Line 70: "....he projects [9-10] consequently...." Please add a full-stop and capital C. "...he projects [9-10]. Consequently..."
It is corrected.
Line 77: "In other study...." to change to "In another study...."
It is corrected.
Line 93 has a different font size than the rest of the paper
It is corrected.
Line 94: "onlines" to be replaced by "outlines"
It is corrected.
Figure 1 should be fully translated into English language "ej? SEC, MINVU???" as well as the central box. Also please revise Figure 2 "ej??" Moreover, Figure 3 has some very very small text that cannot be read clearly. Please enlarge the font slightly.
It is corrected.
Line 205: "...sale of energy..." to be replaced by "...sell energy..."
It is corrected.
Line 215: "planification process" to be replaced by "planning process"
C It is corrected.
Line 220-231: How did the current worldwide energy crisis impact this process? Is it still being re-evaluated every 4 years similar to the business-as-usual scenario?
Yes. Current regulation still evaluates every 4 years.
Line 244: "exist" to be changed to "exists"
It is corrected.
Line 324: "y or region, therefore, t". Please add a full stop and capital T. "y or region. Therefore, t"
It is corrected.
Line 337: What do you mean by "sample universe"?
We meant “the survey participants”
Lines 411-423 identify two separate issues. The first is the cost of the PV system and its associated payback period. The second is the purchasing power of the small-scale investors. These two factors need to be considered separately because they are mutually exclusive. Please revise the paragraph.
Many thanks for pointing it out. We have carefully revised this paragraph and included your suggestions.
Comment on Results: The analysis of results and proposals for mitigation are dealt with within the same paragraphs. May I suggest that a Table is added summarising the proposed mitigation measures for each barrier, before the Conclusion.
Many thanks for your recommendation. We have added a table proposing mitigation measures.
Round 2
Reviewer 2 Report
Accept in present form